# Exploring Factors Contributing to the Implementation of Ontario’s Healthy Kids Community Challenge: Surveys and Key Stakeholder Interviews with Program Providers

**DOI:** 10.3390/ijerph182111108

**Published:** 2021-10-22

**Authors:** Michelle M. Vine, Rachel E. Laxer, Jessica Lee, Daniel W. Harrington, Heather E. Manson

**Affiliations:** 1Ontario Agency for Health Protection and Promotion (OAHPP), Toronto, ON M5G 1M1, Canada; mvine@brocku.ca (M.M.V.); Jessica.lee@oahpp.ca (J.L.); Daniel.harrington@oahpp.ca (D.W.H.); 2Department of Health Sciences, Brock University, St. Catharines, ON L2S 3A1, Canada; 3Dalla Lana School of Public Health, University of Toronto, Toronto, ON M5T 3M7, Canada; heather.manson@utoronto.ca

**Keywords:** health promotion program, child health, community-based program, community partnerships

## Abstract

(1) Background: To explore factors contributing to the Healthy Kids Community Challenge (HKCC) program implementation; (2) Methods: Data were collected through a quantitative survey (n = 124) and in-depth telephone interviews (n = 16) with program providers. Interviews were recorded and transcribed for thematic analysis using NVivo; (3) Results: Provincial funding and in-kind support from community partners were key. Initiatives were feasible to implement, and key messages were well-received by communities. Specific practices and process were commonly discussed, and strong local program leadership was crucial to implementation; (4) Conclusions: Results have implications for planning and implementing future multi-component, community-based health promotion programs that include similar partnerships.

## 1. Introduction

Rising rates of overweight and obesity represent a serious public health issue [1]. Overweight and obesity result from an imbalance between energy consumption and energy expenditure [2]. Overweight is characterized by a body mass index (BMI) of 25, while obesity is characterized by a BMI of 30 or more [2]. In 2018, 26.8% of Canadian adults (aged 18 and older) reported that they were obese, with 36.3% reporting they were overweight [3]. Overweight and obesity prevalence rates in children and youth (aged 3–19 years) decreased (30.7% compared to 27%) between 2004 and 2013 in Canada, with obesity prevalence stabilizing at 13.4% during this period [4]. Boys are more likely to report overweight or obesity than girls and obesity, obesity rates are higher in adolescents than in younger children and low-income population groups are more likely to be at increased risk of obesity [5,6]. Tracking childhood overweight and obesity into adulthood is important given its likelihood of persisting into adulthood, as well as possible long-term consequences of adult weight status [7].

Research indicates that overweight and obesity can significantly impact the physical and psychological health of individuals, including children and youth [8]. Being overweight or obese are risk factors for cardiovascular diseases, type II diabetes, cancers, reduced quality of life and premature death [9]. Research indicates that the etiology of obesity is complex and involves an examination of the role of dietary and lifestyle factors, particularly sugar-sweetened beverages, poor diet quality, prolonged screen time, lack of physical activity, built environment characteristics and short sleep duration or shift work [9]. These environments and modifiable behaviours have been the focus of health-promoting interventions to reduce overweight and obesity and improve health [10]. 

Health promotion involves combinations of educational, organizational, regulatory and political support for environmental and behavioural changes to support health [11]. There is evidence that community-based health promotion programs that intervene at a range of levels (e.g., individual, organizational, community) can have an impact on such health behaviours and outcomes such as overweight and obesity [12,13,14,15,16]. Research suggests that interventions that engage the target population through multiple group activities or activities offered throughout the program duration are most successful in supporting behaviour change [12].

Evidence from interventions to support physical activity and healthy eating, specifically, indicates that (1) physical activity or diet-physical activity interventions delivered in schools with home or community components [13,15]; the adoption of a comprehensive school health approach (i.e., support for improvements in students’ educational outcomes while focusing on health from a holistic, planned and integrated way) [14]; and interventions including physician advice or individual counselling, work activities, mass media campaigns and legislative interventions [16] are most successful in supporting health behaviour change. There is a need for continued evaluation of evidence-based strategies to support health promotion of child health behaviours [17,18,19]. 

### 1.1. Ontario’s Healthy Kids Community Challenge

Drawing on the Ensemble Prévenons l’Obésité Des Enfants’ (EPODE, Together Let’s Prevent Childhood Obesity) model [20], the former Ontario Ministry of Health and Long-Term Care (MOHLTC) developed the Healthy Kids Community Challenge (HKCC). The aim of the HKCC was to improve health behaviours of children aged 0–12 towards addressing high rates of overweight and obesity. Forty-five participating communities were funded by MOHLTC to work with local partners to develop and implement community-based activities (i.e., programs, policies, environmental support) that promoted healthy lifestyles. Each HKCC community hired a local project manager (LPM) for program coordination, planning and leadership, and every LPM invited multi-sectoral community partners to form a Local Steering Committee (LSC) to help plan and implement activities, negotiate private partnerships and offer input on data collection and evaluation [21]. The LPM and LSC (collectively referred to as “local program providers”) and local partners developed and implemented community-based initiatives (programs, policies, environmental support) that promoted healthy lifestyles, based on four themes:*Run. Jump. Play. Every Day*, which promoted physical activity.*Water Does Wonders*, which promoted water consumption and reduced sugar-sweetened beverage consumption.*Choose to Boost Veggies and Fruit*, which promoted fruit and vegetable consumption.*Power Off and Play!* which promoted physical activity and play in place of screen time and sedentary behaviour.

MOHLTC supported HKCC communities through funding, training, technical support, social marketing tools and other resources. Public Health Ontario (PHO) undertook a provincial evaluation of the HKCC to assess the extent to which its goal and related objectives were achieved over the three-year intervention period (2015 to 2018). While 45 communities across Ontario were selected to participate in the HKCC, this study is focused on the 39 municipally-funded communities as there is a separate community-partnered evaluation to assess the implementation and impact of the program as implemented by six Aboriginal Health Access Centres or Aboriginal Community Health Centres in Ontario.

### 1.2. Theoretical Framework

Our process evaluation of the HKCC drew on Durlak and DuPre’s Ecological Framework for Understanding Effective Implementation [22]. Durlak and DuPre’s framework was utilized to help explore factors that contributed to successful program implementation, using a multi-level ecological perspective [22] (see Figure 1). This framework was used to guide data collection and analysis tools to understand how individual and organizational capacity for HKCC implementation was affected by the interaction of a range of factors operating across five categories. HKCC program characteristics were mapped onto the five categories, as follows:Community level factors: How the community and/or provincial context relates to HKCC program implementation (i.e., research systems, politics, funding).Provider characteristics: Local program providers’ perceptions of the need for and potential benefits of the program, as well as perceived self-efficacy and skill proficiency for the desired tasks.Innovation (Program) characteristics: Characteristics of HKCC programs and initiatives at the community level (i.e., adaptability and compatibility, contextual appropriateness).Organizational capacity (Prevention delivery system): Factors related to the system delivering the HKCC. The LSC has a key role in planning and implementing initiatives in communities, and consideration of the organizational features, practices, processes and staffing of this group is relevant.Training and technical assistance (Prevention support system): training and effectively preparing local program providers to complete expected tasks related to the HKCC (i.e., active training and ongoing resources).

## 2. Methods

The objective of the process evaluation was to explore factors contributing to implementation of Ontario’s HKCC and to assess local program providers’ knowledge, attitudes and behaviours as they related to the program and its implementation. The evaluation utilized a mixed method design, including data from an online survey and in-depth semi-structured telephone interviews, both collected in 2018 near the program’s end. This manuscript integrates key findings from both data sets to explore factors contributing to implementation of the HKCC over the course of the three-year program (2015–2018).

### 2.1. Data Collection

#### 2.1.1. Local Program Provider Survey

LSC survey data were collected at two time periods (Time 1 data are reported in [23]). This manuscript reports on Time 2 survey data (n = 124), which were completed online in August 2018 by local program providers using an online survey tool. The survey was piloted in one community (n = 8; 1 LPM, 7 LSC members), and some of the original survey questions were modified based on participant feedback prior to sending the survey to the remaining communities.

Drawing from Durlak and DuPre’s framework, the LSC survey asked participants about their perceptions of program implementation, including the following: barriers and facilitators; roles within and attitudes towards the HKCC; health equity and reach; local program providers’ functioning and relationships among partners; training and other support.

#### 2.1.2. Local Program Providers Interviews

Semi-structured telephone interviews (n = 16) were undertaken with LPMs and LSC members from eight HKCC communities between August and October 2018 (Time 2; Time 1 interview data are reported elsewhere in an internal report [24].

An interview guide was developed based on previous evaluation activities, Durlak and DuPre’s framework [22] and a review of literature on effective implementation of community-based interventions. Similar to the approach taken with the LSC survey, the interviewers asked local program providers about their role and contributions to the HKCC program; perceptions of the need for and benefit of the HKCC; insights about the reach and adoption of the HKCC; perceptions about the program’s implementation (e.g., resources and support, stakeholders, partnerships, social marketing); political commitment, community outcomes related to the HKCC and long-term program sustainability. The interviews were audio recorded and transcribed verbatim for thematic analysis using NVivo 11.0 software (QSR).

#### 2.1.3. Participant Recruitment

A fan-out sampling approach was used to recruit local program providers to participate in the survey via an email that included instructions and a survey link. The survey was initially shared on 7 August 2018, with four email reminders following, before it closed on 12 September 2018.

The survey was completed online by local program providers (both LPMs and LSC members) using Surveys@PHO, an online survey platform hosted by PHO. Based on pilot feedback, some of the original survey questions were modified prior to sending the survey broadly to the remaining communities.

Interview recruitment occurred via email invitation, and LSC members were asked to communicate their interest to PHO by email or telephone. Participation in the survey and interviews was voluntary. Participants provided written informed consent to participate in this evaluation. Ethics approval for the survey and interview protocols was received from Public Health Ontario’s Ethics Research Board in August 2015.

#### 2.1.4. Data Analysis

Survey analysis includes descriptive statistics, which were analyzed using Microsoft Excel. Data were grouped into three categories: agree/strongly agree, neither agree nor disagree and disagree/strongly disagree. In some instances, an “other” category was reported, which encompassed disagree, strongly disagree, prefer not to answer and don’t know.

For the interview data, a theme code set was developed deductively according to the constructs of the Ecological Framework for Understanding Effective Implementation [22] and inductively based on a preliminary review of four interview transcripts. Sixteen interviews were transcribed verbatim, with written permission, for subsequent analysis. Interview transcripts were coded and analyzed by a member of the PHO evaluation team using NVivo 11.0 (QSR), where passages of text were selected that aligned with codes included in the theme code set.

Inter-rater reliability was conducted by two coders using a sample of interview transcripts (10%) to determine the reliability of the theme code set and to resolve differences in interpretation. A kappa coefficient was calculated to determine the degree of agreement in coding. A Kappa score of 0.62 was calculated, which represents a substantial amount of agreement [25,26]. Discrepancies in coding were reconciled by discussion and clarification of themes.

Factors that contributed to the HKCC’s implementation are presented according to the five levels outlined by Durlak and DuPre’s framework [22]: community-level, provider-level, program-level, organizational capacity and training and technical assistance. Findings related to the perceived sustainability of the HKCC are also integrated into the findings.

Quantitative survey data are presented as descriptive statistics, including open-ended data that are included to augment the survey data. Qualitative interview data are presented as verbatim quotations according to the number of times they were mentioned (i.e., mentions) and number of sources (i.e., participant interviews) in which they were mentioned.

## 3. Results

### 3.1. Participant Characteristics

Overall, 124 local program providers from 26 HKCC communities completed at least a portion of the survey, with 27% identifying as an LPM (four communities had two participating LPMs; one community had three participating LPMs) and 73% identifying as an LSC member. Most LPMs (74.2%) had been in their role for more than one year, while most LSC members had been involved in the HKCC for two or more years. LPMs were employed in a range of sectors, including public health, non-profit sector and local municipalities. Almost all LSC members who completed the survey had been involved in the HKCC for themes three and four (i.e., during the second half of the program).

Sixteen local program providers from ten HKCC communities participated in an interview. Participants consisted of two LPMS and 14 LSC members, also employed in a range of sectors.

### 3.2. Community Level Factors

Community-level factors refer to community and/or provincial contexts with the potential to affect implementation of the HKCC, including prevention theory and research, political support, funding and/or in-kind support from partnerships, policy support and community context.

All 16 interview participants discussed the role of community-level factors in local HKCC implementation (see Table 1). All participants discussed the role of their community’s structure (e.g., physical and socio-cultural elements) in shaping the implementation of the HKCC (mention = 61, interview = 16). While the structure of the community does not appear in Durlak and DuPre’s Framework [22], it is essential to understand its role in shaping implementation given that the HKCC program was delivered locally within geographically-defined communities.

The community structure for HKCC programming was discussed in relation to local geography (e.g., municipal boundaries, physical boundaries, urban–rural differences) and local population (e.g., socio-economic status, language, culture, population size and/or sub-groups). For example, some communities used municipal boundaries to facilitate program planning:


*“[W]e divided [our community] into five sections. [Our community] is a natural…like there’s a part that sticks out…so that was one section, and then [we] took the rest of the city and divided it in quarters and looked at that as local planning groups.”*
—ID 19

Participants indicated the extent to which scientific theory and research were integrated into their planning and implementation of HKCC interventions. Both interview and survey respondents indicated that evidence supporting HKCC initiatives was strong (mentions = 34, interviews = 14 (Table 1), 98.2%, respectively).

Survey and interview results both indicated that HKCC themes were evidence-based. Most survey respondents observed that the HKCC’s four program themes were based on sound scientific evidence (92.8%) and that the health risks and benefits of the population had been considered in program design (91.9%). The integration of available scientific evidence into the design of local HKCC initiatives was also demonstrated:


*“[T]hey’d have the physical activity specialist kind of give some evidence and then the program leads, or the project leads, would have that information for when they designed their projects, which kind of try and align with what HKCC was trying to do and with what evidence for that theme was available.”*
—Interview ID 8

Local community partner support for HKCC implementation was perceived to be high. Most (91%) survey respondents felt strong support from local community partners, while less than half (47.6%) indicated that support for the HKCC came in the form of local political commitment. Several interview participants (mentions = 16, interviews = 11) indicated that their mayors and local municipalities provided meaningful support to local HKCC programs. Support ranged from the endorsement of the local mayor to municipalities leveraging their resources and infrastructure for the LSC to help move HKCC programs forward. For example:


*“[O]ur champion was our Mayor and he bought into this program 100%. He attended almost every activity that happened.”*
—Interview ID 4


*“[H]aving our individual mayors and councils support made it [so] that we could move our initiatives forward. And, to have municipal resources at our disposal when we needed things like administration support, tech support, those types of things, we were able to access them through our individual municipalities.”*
—Interview ID 1

Despite reports of meaningful local political support for communities, four interview participants described that it could have been stronger and more sincere. As one participant described:


*“[T]he municipality led the work, and the mayor was the champion. And I’m going to say that the political commitment was a little bit superficial. And just that it didn’t take full advantage of what that could have meant for being a visible leader.”*
—Interview ID 17

Program providers indicated that program funding and in-kind support were essential components of HKCC development and implementation. Almost all interview participants (mentions = 20, interviews = 14) acknowledged funding as valuable, while just over half (55.9%) of survey respondents reported that a sufficient level of funding was in place from the province to support HKCC implementation. In particular, funding and in-kind support were key elements to help facilitate the engagement of community partners and development of HKCC programs. For example:


*“[F]rom our perspective I do think because there was some funds that came along with that we were able to do a lot more things to reach the community and engage people than we would have otherwise, because everybody has their own limited budgets with their strategies and what they already do.”*
—Interview ID 14

Eighteen (66.7%) LPM and 23 (32.4%) LSC member survey respondents indicated that their HKCC community had received additional funding from their host organization (e.g., public health unit, recreation centre, etc.) to support HKCC programming. In-kind support from host organizations was provided to 74.6% of program providers, including, for example, staff expertise and marketing support, use of facilities, office equipment, programming and administration. 

### 3.3. Provider Characteristics

Provider characteristics refer to local program provider perceptions of the need for and potential benefits of the HKCC program—an important determinant of successful implementation—and their perceived self-efficacy and skill proficiency (e.g., whether local program providers felt they had the skills required to support HKCC implementation).

Local program providers reported that they perceived a local need for the HKCC, and that the program was highly beneficial to their communities. For example, survey data indicated that most (90%) providers agreed or strongly agreed that the HKCC was effective in increasing knowledge of health behaviours, that the HKCC program was beneficial in supporting child health behaviours, that there was a need for HKCC initiatives in communities, that the HKCC enhanced child and parent access to programs and activities and that it contributed to a sense of community. In the same way, nearly all interview participants indicated that the HKCC program was beneficial to their community (mentions = 78, interviews = 15). One reported benefit included changes in the health behaviours of children and families (e.g., reduced screen time, increased in water consumption):


*“I have participated in something [called] Walking to School Wednesdays at one of the local schools…I stand there and I see all these kids with their water bottles. Every single kid comes along with a water bottle. Like, that’s quite a revolution.”*
—Interview ID 19

Among survey respondents, more than half agreed that the HKCC was effective in changing health behaviours (64.3% agreed/strongly agreed); however, there was less agreement that the HKCC was effective in reducing childhood overweight and obesity (46.8% agreed/strongly agreed). 

Most survey participants who were confident in their ability to engage partners (93.6%) had the belief that their community could achieve the goals of the HKCC (92.7%) and had the self-confidence and ability to implement the program in their communities (89.9%). Examples of strategies for engaging HKCC partners included targeting partners working in settings where children live, work and play (e.g., schools, daycares, libraries, etc.), leveraging existing relationships through the themes and sharing HKCC information via social media platforms and on websites.

LPM skills related to management and coordination of the HKCC were noted as vital components of successful program delivery and coordination of LSC member committees (mentions = 37, interviews = 14). Participants also reported an alignment between the LPM’s professional background and the knowledge and skills required to coordinate the HKCC program.

Finally, the content expertise (e.g., children’s health, physical activity) of LSC members was observed to be beneficial in relation to planning HKCC programs in alignment with theme-based messaging. For example,


*“I started out in childcare and then advanced to the current position of director, so I’ve always had a very keen interest in the wellbeing of children and their families in our municipality. When our municipality [applied] to partner in this initiative with the other…communities [in our region], I right away thought that would be a great fit if I became a member of the committee.”*
—Interview ID 15

### 3.4. Innovation Characteristics

Innovation refers to the characteristics of the HKCC program that influenced its implementation. These characteristics as reported by interview participants are outlined in Table 2 and described in detail below. They include partnerships, the focus of HKCC initiatives, social media and social marketing, adaptability, compatibility and program reach. Perspectives on and plans for sustainability of the program are incorporated into this section.

#### 3.4.1. Partnerships

All communities were encouraged to form and sustain multi-sectoral partnerships to support planning and implementation of the HKCC. Almost all program provider survey respondents reported that local community partners expanded their own programs to include HKCC initiatives (90.9%) and to spread key messages about the HKCC into the community (97.5%). Eighty-one percent of survey respondents reported having networked with diverse sectors to gain HKCC support, while new community structures and networks were developed by 73.5% of respondents.

In both surveys and interviews, local community partnerships were the most commonly mentioned characteristic that affected the implementation of the HKCC in both surveys and interviews. These partnerships were largely discussed by interview participants as a success (mentions = 116, interviews = 16). There were a variety of partnerships with local organizations and community members that LSCs leveraged to support delivery of programming. Both private (e.g., grocery stores) and public (e.g., recreation settings) organizations partnered with HKCC communities to provide the setting for HKCC initiatives, highlighting the critical importance of partners in program planning and delivery. 

While the success of partnerships was a key theme that emerged from the interviews, many participants discussed challenges experienced in the development and maintenance of partnerships (mentions = 32, interviews = 15). These challenges originated from the limited time and resources of local organizations that the LSC had an interest in partnering with: *“I think there might have been some collaborative efforts to work a little bit together. But again, you can always do more but you’re sort of limited by time, you’re limited by resources, and then you have to get your own work done, too”* (ID 6). Many interview participants reported difficulties engaging schools and school boards in their local HKCC planning and implementation (mentions = 11, interviews = 9):


*“It was a little bit difficult to get the right representation from the education field…that’s just because of the nature of their beast, right. Our meetings were happening during the day, so, you know, your champion teacher wouldn’t be available…and then if you went into administration, it was hard to get us on the top of the list for a meeting to attend.”*
—Interview ID 4

Most survey respondents indicated that coordination among partners had improved over the course of the program, and that there was a shared HKCC-related vision and goals among partners. Fewer, but still the majority of respondents, thought that their established partnerships would be sustained after the funding ended and that trust among partners had increased throughout the HKCC.

#### 3.4.2. Focus of HKCC Programs

HKCC programs reflected the themes developed by the MOHLTC, which focused on increasing physical activity, water consumption, healthy eating and decreasing screen time. There are also examples of programs that were adapted or they changed the key messages provided by MOHLTC to promote the theme. One community modified their theme two tagline to say “*Water is the way to Go!*” to emphasize the fact that the healthy choice and action they were trying to influence was to choose water. In addition to the theme three “*Choose to boost veggies and fruit*” tagline provided by MOHLTC, another community also included the tagline “*Powered by veggies and fruit*” to appeal to a different age groups/audiences and tie in recreation/sports as fruits and vegetables provide the fuel and power to be active and participate in sports. Adaptations to theme 4 also emerged, as one community indicated that they used the same message for most of the program, with occasional adaptations to focus on land-based activities and the well-being that land-based activities bring to a community. The same community also adapted their messaging to highlight the importance of unstructured play for children.

#### 3.4.3. Social Media and Social Marketing

There were a variety of strategies reported to promote key messages of the HKCC, including social media, local newspapers, radio stations and print advertising (e.g., flyers). For example, some communities leveraged pre-existing social media platforms run by local partners in order to reach an established social media following, e.g.,


*“Our Healthy Kids social media…has been linked to the [regional public health unit’s] social media platforms…sometimes it can take years to gain a following and so, we wanted to link on to… another organizations platform that has the following.”*
—Interview ID 11

Nineteen (70.4%) LPMs indicated that they used more than one type of social media platform to share information about the HKCC. Facebook and Twitter were the most commonly used platforms, given that they can be used to share both messages and photos. Approximately half of the LPMs indicated that they leveraged another social media account of relevance in their community to help spread their message, either in addition to, or in place of, their own HKCC accounts (e.g., HKCC parents’ or LSC member personal accounts, school boards promoted programming via their calendars or social media accounts).

MOHLTC provided LPMs within HKCC communities with social marketing materials to assist in the delivery of the HKCC. Survey results indicate that most of the communities either used the materials as provided (37%) or made necessary adaptations to better fit their context (52%).

#### 3.4.4. Adaptability and Compatibility

Program adaptability and compatibility are recognized as important mechanisms for tailoring programs to specific stakeholders and audiences [27]. Adaptability refers to the degree to which a program can be modified to fit the needs of local providers, and compatibility refers to the fit between the program and the local context [27].

Given the range of communities participating in the HKCC program and generic messaging and marketing materials provided by MOHLTC, some interventions were adapted by program providers in order to meet the needs of local communities. Almost all interview participants noted that they were able to adapt the HKCC program (mentions = 35, interviews = 13; Table 3). This was often discussed alongside the ability to reach the local population:


*“[Rural community members] shop in their own communities in their small markets. So a way of… overcoming this barrier…is we designed our own nutrition program… with our Public Health registered dietician … [and] we partnered with local grocery stores in our townships and delivered the programming that way…We just realized through the process that…one size doesn’t always fit all… [and] we have to come up with creative ways to making sure we’re reaching our rural communities, as well.”*
—Interview ID 11

Compatibility of HKCC programming with existing community priorities and infrastructure to promote health behaviours was a key element of successful program implementation and was discussed by most interview participants (mentions = 34, interviews = 13). There was also agreement that the HKCC was adaptable to community needs (98.2% of survey respondents), initiatives were feasible for communities to implement (91.0%) and that key messages were positively received by communities (88.3%).

Interview participants discussed the HKCC’s compatibility with the priorities and practices of local organizations, specifically those that already serve children and families living in HKCC communities. The compatibility between local partners and infrastructure helped the LSC to leverage and support existing community programs. For example, 


*“The [XXX] Child and Needs Initiative and they have a critical hours group, which is called Growing up Great and they also are looking at um, child health behaviours and um, programs and training that support that as well and they were one of our partners as well. This is one of the reasons why we reached out to them was to leverage what they were working on and see how we could work together to uh, to strengthen that.”*
—Interview ID 1

#### 3.4.5. Reach

Most survey respondents (93.7%) described a focus on reaching populations within their communities that faced structural barriers to health based on race, ethnicity, indigeneity, socioeconomic position and rurality, among other factors. Many respondents (41.8%) described challenges in reaching specific groups, citing, e.g., the time it takes to engage in the absence of pre-existing relationships and the impact of changes to organizational staff or funding among HKCC partnersKCC. Participant-level barriers to reach were reported to include cost (e.g., of healthy food on a limited budget—although most program activities were free or subsidized for participants), accessibility (e.g., language barriers) and perceived lack of interest.

Geography and location were identified as key challenges to reaching certain local populations, as some rural communities that participated in the program did not have centrally located hubs (i.e., libraries, community meeting spaces) where people could gather. One strategy identified for reaching children in HKCC communities was to deliver programming in settings that children already access:


*“[W]e kind of went to where they were…I think that was the main way [of reaching children], going to the schools, going to community events, rec centers… and sports associations…places where kids would be.”*
—Interview ID 10

While schools were perceived as an appropriate place to reach children, some communities were challenged by a lack of school-level capacity for program participation. For example: *“[S]ome schools’ participation was lower than we wanted, but I don’t really think that was HKCC related, it was kind of school by school and the enthusiasm of each school and their willingness to participate”* (Interview ID 8). These opposing experiences reaching community members through the school setting highlight the important role that local context can have for local partners in HKCC implementation.

#### 3.4.6. Sustainability

While sustainability is not a construct of Durlak and DuPre’s Framework, it aligns with the purpose of the Framework, which is to focus on evaluating the capacity to identify, select, plan, implement, evaluate and sustain interventions [22]. Asking respondents about their perception of the sustainability of the HKCC in their community lends itself to understanding and evaluating the local capacity to sustain the program.

Survey and interview results indicated that sustainability of the HKCC program was an important implementation outcome. Sustainability can be measured through program maintenance or the capacity to continue to deliver a program through a network of agencies, instead of, or in addition to, the agency that originally implemented the program [28]. More than half of survey respondents indicated that there was potential for the HKCC to be sustained beyond the funding period. At the time of the interviews, some participants discussed actions that were taken to ensure that HKCC programs would still be accessible to the community after the program had ended. These efforts usually involved working with community partners to embed program materials into other local programs and settings:


*“Healthy Kids superhero toolkits which includes all of the resources from Themes 1–4 and it also includes, a new live theatre script that incorporates all four themes…and it was videotaped so that it can be accessed by schools and other community groups in the future.”*
—Interview ID 11

Other participants noted that toward the end of the program, when the LSC interviews took place, they were engaging in discussions on how to sustain theme-based messaging and programming that was implemented over the course of the program:


*“Well, that’s about where we are at this moment is trying to solidify that by talking to our partner agencies. We’re having conversations with our local health unit. We’re having some conversations at the municipal level and then we will be having them at the school board level to find ways with the resources that we have at our disposal to keep things moving forward.”*
—Interview ID 15

Nearly 75% of survey participants recognized that there were community champions who strongly supported the program, that community members (i.e., parents, children, caregivers) were feeling engaged with the program and that the HKCC was integrated into partner organizations. More than 62% of respondents indicated that their HKCC community had developed a sustainability plan for their program to extend beyond the formal funding period (2015–2018).

There were also reported challenges and barriers to sustaining local HKCC programs. Participants noted that sustaining HKCC programs would be difficult without an LPM or ongoing funding:


*“I think anything that was, sort of special that came out of HKCC won’t happen just due to lack of funding. You know if they had $3,000 for a certain after school program, well, without that $3000 dollars it might not happen, you know?”*
—Interview ID 6


*“I hate seeing [programs] start for two years, two or three year programs and then get cut just right off, and then without some of that funding, the programs do die.”*
—Interview ID 15


*“And I guess my fear would be, without the Healthy Kids [LPM] in the future it’s going to be very difficult… as much as we’re going to try to transfer and keep it going, and what not, I think that’s going to be really hard for those partnerships, to survive when there isn’t the [LPM] there to, kind of, be that glue to hold it all together.”*
—Interview ID 16

### 3.5. Prevention Delivery System—Organizational Capacity

Organizational capacity to deliver the HKCC program refers to the functioning of the local program provider (general factors such as organizational climate and shared vision, specific practices and processes, specific staffing considerations), and relationships among HKCC partners. Organizational factors were essential to the successful delivery of the program.

Across the 16 interviews, specific practices and processes engaged in by LSCs (i.e., communication, coordination with other agencies, etc.) were the most commonly discussed factors that affected their organizational capacity to implement the HKCC (Table 3). There was general agreement that specific staffing considerations, including a successful LPM, were crucial to the successful implementation of the HKCC.

Positive organizational climate and the presence of a shared vision were two general organizational factors that supported an LSC’s ability to plan, develop and implement the HKCC program (Table 3). Shared decision-making helped to ensure that the HKCC program matched the local needs, contexts and preferences of community members and organizations operating within each HKCC community. For example, one participant credited the collaboration between LSC members and the organizations they belong to as a key factor helping to support implementation of HKCC initiatives as they were planned:


*“I think pretty much what the plan was, we were able to follow through with it and I think a lot of that resulted from the fact that our committee was a collaboration of a number of municipalities and communities. We would have never been able to achieve what we did if it was a single community trying to carry the initiative forward, so the fact that it was a collaboration of many communities, that gave us the power and support that we needed to move things forward.”*
—Interview ID 15

There was agreement that communication between and coordination of LSC members helped to facilitate successful program execution. In terms of communication, LSC members discussed the purpose and function of their LSC meetings and how they contributed to planning their local HKCC programs. Other interview respondents discussed the role that LSC meetings played in strengthening connections between LSC members and the organizations they were employed by. For example:


*“I think the regular meetings helped, as well because it keeps people engaged, inspired, and it keeps those connections going with those partnerships”*
—Interview ID 6

Results from interviews and survey respondents are aligned, where more than 80% of LSC members agreed that there was strong LSC leadership, that community champions supported the HKCC and that they were satisfied with the way the LSC functioned, including good communication among LSC members (see Figure 2).

There was general agreement that local program staffing considerations were crucial to the program’s success (mentions = 49, interviews = 15). Participants discussed the importance of the leadership and support of LPMs, LSC members, program champions and managers within host organizations for the HKCC’s success. Interview participants mainly discussed the leadership of their LPMs and LSC members as being vital to local HKCC implementation (mentions = 30, interviews= 13). For example:


*“I think that having [an LPM], particularly in our situation because it was multiple rural communities, having someone to tie everybody together and a little bit of the planning and promotion, and rolling things out, I think that for us, was huge.”*
—Interview ID 14

### 3.6. Prevention Support System—Training and Technical Assistance

Training refers to support that prepared providers for their new roles related to the HKCC, while technical assistance refers to support provided to communities after implementation commenced. There was agreement among LPMs who participated in the survey that the MOHLTC and other resources (i.e., Healthy Kids Resource Centres) were helpful throughout implementation of the HKCC.

While few interview participants discussed the importance of these factors, some participants discussed the value of technical assistance provided to their community by the MOHLTC (mentions = 9, interviews = 6):


*“I think the resources that were created by the Ministry were essential to the success, that Ministry support was essential to the success.”*
—Interview ID 13

Some interview participants noted areas where the technical assistance from the MOHLTC could have been improved. For example, a delay in receiving materials and messaging acted as a barrier to program implementation:


*“What was hard, though, was the fact that they delayed, at the Ministry level, the pouring out of the themes, okay. Your next theme is water and we’ll get that material to you when we do and we’re all standing by waiting for stuff and nothing happens because something stalled at the Ministry.”*
—Interview ID 19

## 4. Discussion & Conclusions

The objective of this study was to explore factors contributing to implementation of the HKCC—a province-wide community-based program intended to improve children’s health behaviours in Ontario. Findings from the survey and interviews of LSC members and LPMs in HKCC communities indicate that Durlak and DuPre’s [22] framework can be used as a guide to identify factors contributing to program implementation. Findings can provide policy makers and practitioners with important information to support intervention replication, including knowledge about how to implement complex interventions [29].

Based on the EPODE model [27], the HKCC program was based on sound scientific evidence. The HKCC program integrated scientific evidence through a review of related literature to consider health risks, application of theory [22], collection of new data through evaluation and the involvement of experts, stakeholders (i.e., LPM, LSC members) and community members in program planning. While the approach has some strengths, planning for health promotion intervention development, implementation and evaluation can follow a more systematic process and protocol, including, for example, intervention mapping.

Provider-level factors include the belief that the HKCC was both needed and beneficial and that it contributed to a sense of community and enhanced access to programming. However, among survey respondents, more than half agreed that the HKCC was effective in changing health behaviours; however, there was less agreement that the HKCC was effective in reducing childhood overweight and obesity. These findings are not surprising, given the downstream nature of these outcomes and the focus of the program on the promotion of healthy behaviours.

Regarding organizational capacity, program staff members were critical to successful program implementation, including strong leadership from the LSC, and that community champions can play a key role. There was satisfaction with the LSC functioning, including a communication mechanism between members. LPMs and LSC members were confident in their ability to engage partners and achieve program goals, and LPM management and coordination was vital to program delivery and coordination of the LSC. Hiring and managing high-quality staff to run health promotion programs is noted as a key leadership function [30]. Specifically, hiring managers should consider whether potential employees have a skill-set and experiences that are matched to the program goals, have interpersonal qualities need for community engagement, display cultural competence to engage in supportive relationships with stakeholders, participants and staff members and have an interest in the host organization’s mission [30].

Stakeholder relations, interpersonal skills and cultural competence can help to support partner development and targeted reach of the HKCC. Targeting settings (i.e., physical location) where children and parents live, work and play helped to improve intervention reach. However, there may be further opportunity to conceptualize and assess the context and implementation of this complex intervention more fully. For example, Pfadenhauer, Gerhardus, Mozygemba, Lysdahl, Booth, Hofmann et al. [31] developed the Context and Implementation of Complex Interventions framework that could be a useful tool to assess the integration of context, implementation and setting to advance understandings of how the HKCC worked.

While most communities focused on reaching populations that faced structural barriers to health, some experienced barriers in doing so, including a lack of time allotted to develop relationships with local populations, turnover of HKCC program staff, funding and challenges related to geography and transportation to access programming. Participant-level barriers included accessibility (e.g., language of program delivery) and cost (e.g., of healthy food while on a limited budget). If similar future community-based health promotion programs aim to reduce health disparities, they should use a structural approach to target upstream determinants of health and attempt to strike a balance between individual-focused and structurally focused health promotion interventions [32].

Some notable challenges were evident with partner development. Previous research examining multisectoral partnership development in health promotion identified core elements of positive partnership processes including shared mission, diverse partners, trustworthy leadership, balance formal and informal roles, trust-building throughout, balance maintenance and production activities, consideration of the context (social, cultural, economic) of the work and evaluation of the partnerships for consistent improvement [33]. These elements may be considered to guide the development of future partnerships in similar community-based health promotion work.

Previous research highlighted the need for politicians to act as advocates of programs at local and provincial levels, in addition to mobilizing their peers to modify programs and policies that support physical activity and healthy eating environments [27]. Relatedly, building networks with local partners, including local companies, retailers and supermarkets, can lead to collective action and goal accomplishment, financial resources, relevant skills or in-kind resources (e.g., equipment) [27]. Future research exploring the implications of variable political support for population-level social marketing interventions, as well as strategies to improve the feasibility of forming local private partnerships, is needed. These strategies include, but are not limited to, building trust, discussion about conflicts of interest and ethics, monitoring and evaluation and good governance [34].

### Implications

These evaluation results may benefit communities directly (e.g., local project managers, local steering committees, other key stakeholders) given that data may inform future training and support, and therefore, increase local-level community capacity for delivering community-based health promotion programs, activities and services. The findings may have positive implications for the local level policy (e.g., school nutrition/physical activity), and urban planning initiatives (e.g., built environment: walkability, public park design, bike lanes).

Broadly, the findings support the value in monitoring intervention adaptation and compatibility to understand how the HKCC worked in real-world settings. Specifically, Durlak [35] identified the need to evaluate the impact of adaptations and to ascertain which components of implementation (e.g., tag line, program components, delivery setting) are associated with program outcomes. While fidelity and program dosage are often the focus of research on program implementation and outcomes, a focus on program adaptations that occur during program implementation to assess the extent to which they impact or influence different outcomes (e.g., quality of program delivery, participant responsiveness) [35] is also warranted.

Strategies to successfully engage and reach populations that face structural barriers to health deserve further attention, particularly given the focus of the HKCC on reaching these groups. Future research exploring the implications of political support for population-level social marketing interventions, as well as strategies to improve the feasibility of forming successful local multi-sectoral partnerships, is needed.

## 5. Strengths & Limitations

### 5.1. Strengths

This research has three main strengths. First, the survey and interviews were designed around the Durlak and DuPre framework [22] a priori. The questions directly probed key constructs within the framework that are known to be important to implementation of community-based initiatives. Second, reliability of data analysis was increased by maintaining meticulous records of interviews and through documentation of data analysis, in addition to undertaking an inter-rater reliability exercise [36]. Both content and structure were assessed by two research team members, with a substantial level of agreement (62%), indicating analytic reliability. Third, combining qualitative thematic analysis with quantitative summary of results helped to minimize researcher bias in the presentation of results [37]. Quantifying results into a summary table condenses results for ease of interpretation, while presentation of quotations offers context and rich description [37].

### 5.2. Limitations

In addition to the strengths, the survey and interview data and analysis have some limitations that must been considered. First, the survey and interview recruitment approach—requesting that LPMs forward the survey link to their LSC members—led to uncertainty about how many people were reached as potential participants. This approach prohibited us from calculating an appropriate denominator to report a survey response rate. Second, based on the fact that survey participation was optional, there was a risk of selection bias in those that did not complete the survey. It is also possible that LPMs and LSC members, who were more engaged in the HKCC closer to the end of the program, were more likely to have completed the survey. Third, some survey respondents dropped out as the survey progressed, which could have been as a result of the survey length and the time it took to complete. Finally, these results are not necessarily representative of all LPM and LSC members, given that participation only represented two-thirds of 39 municipally-funded communities. Despite these limitations, the data provide an understanding of the functioning of the LSC and some of the successes and challenges of HKCC implementation.

In the context of the interview data, qualitative research captures stories to understand people’s perspectives, understand how systems function and their consequences for people’s lives, as well as the context—what is going on around the people, groups, organizations, communities or systems of interest [38]. Given the nature of qualitative research, including its small sample size (n = 16), it is not possible to make generalizations across all 39 HKCC communities; however, it is expected that the findings would be similar across HKCC communities and transferable to other (i.e., EPODE-type) programs in contexts similar to those in Ontario. Transferability refers to the degree to which the findings might fit or be congruent between community contexts [38]. Second, data in the interview transcripts were limited to responses the participants provided to questions asked. Given the semi-structured nature of the interviews and individual experiences of participants, some questions elicited more fulsome discussion on a specific topic. In the analysis, there was an absence of certain constructs from Durlak and DuPre’s [22] Framework (e.g., training and technical assistance). However, this may be a result of a lack of probing of these constructs by the interviewer.

## Figures and Tables

**Figure 1 ijerph-18-11108-f001:**
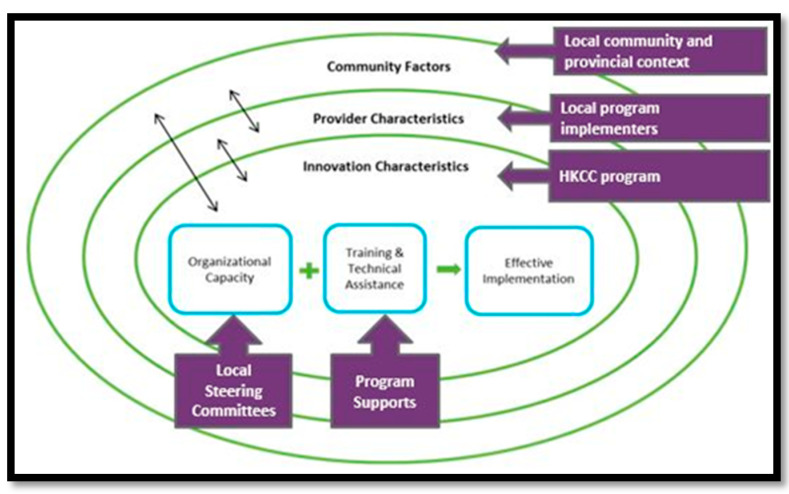
Adapted from Durlak & DuPre’s Ecological Framework for Understanding Effective Implementation (2008) [22].

**Figure 2 ijerph-18-11108-f002:**
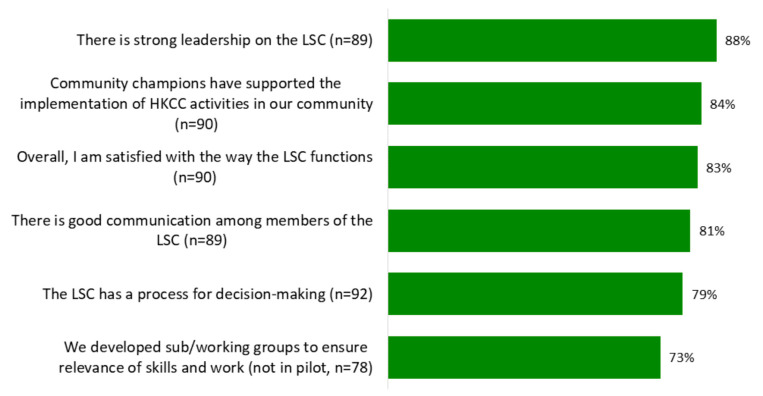
Understanding the local steering committee, as reported by survey respondents.

**Table 1 ijerph-18-11108-t001:** Community-level factors that affected Healthy Kids Community Challenge (HKCC) implementation, reported by interview participants.

Factor	Mention (# of Times the Theme Was Mentioned)	Interview (# of Interviews)
Structure of community	61	16
Prevention theory and research	34	14
Politics	20	15
Funding and support	20	14
Policy	7	4

**Table 2 ijerph-18-11108-t002:** Characteristics of the Healthy Kids Community Challenge (HKCC) program, as reported by interview participants.

Characteristics of the HKCC	Mention (# of Times the Theme Was Mentioned)	Interview (# of Interviews)
Partnerships	148	16
Focus of HKCC initiatives	107	16
Social media and social marketing	46	16
Adaptability	35	13
Compatibility	34	13
Reach of the HKCC initiatives	33	16

**Table 3 ijerph-18-11108-t003:** Factors related to organizational capacity, as reported by interview participants.

Characteristics of the HKCC	Mention (# of Times the Theme Was Mentioned)	Interview (# of Interviews)
A. General organizational factors	27	11
i. Organizational climate	15	8
ii. Shared vision	11	7
B. Specific practices and processes	101	16
i. Communication	31	12
ii. Coordination with other agencies	32	14
iii. Formulation of tasks	27	12
iv. Shared decision-making	15	8
C. Specific staffing considerations	49	15
i. Leadership	30	13
ii. Program champion	14	8
iii. Managerial, supervisory support	5	5

## Data Availability

The datasets used and/or analysed during the current study are available from the corresponding author on reasonable request.

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
