# Peer review of "Exploring Factors Contributing to the Implementation of Ontario’s Healthy Kids Community Challenge: Surveys and Key Stakeholder Interviews with Program Providers"

_ijerph, 2021, doi:10.3390/ijerph182111108_

Round 1

Reviewer 1 Report

Introduction: A more in-depth review of the literature is needed, not only of the structure of the participating communities, but also of the problem at hand and the current status of the issue.

Methods: Some information of the "results" should be in the methods part. It is not clear total sample, maybe a table could make it clear.

The part "Discussion & conclusions" should be before than "Strengths & limitations"

Line651 should be revised and find anothe way to say this ("While partner development was largely successful, some notable challenges were ev- 651
ident").

Author Response

Reviewer #1 Feedback

Author Response

Page #

Introduction: A more in-depth review of the literature is needed, not only of the structure of the participating communities, but also of the problem at hand and the current status of the issue.

Thank you for this feedback. We have spent some time adding to the introduction section in order to highlight the issue of obesity, its causes and determinants, and need for the development of interventions to focus on modifiable behaviour change.

Introduction section, Lines 25-68, pages 1-2.

Methods: Some information of the "results" should be in the methods part.

Thank you for this feedback. We have moved the following text to the methods section (from the results section):

Factors that contributed to the HKCC’s implementation are presented according to the five levels outlined by Durlak and DuPre’s framework [9]: community-level, provider-level, program-level, organizational capacity, and training and technical assistance. Findings related to the perceived sustainability of the HKCC are also integrated into the findings.

Quantitative survey data are presented as descriptive statistics, including, open-ended data that are included to augment the survey data. Qualitative interview data are presented as verbatim quotations according to the number of times they were mentioned (i.e., mentions), number of sources (i.e., participant interviews) in which they were mentioned.

Lines 292-302, page 4.

It is not clear total sample, maybe a table could make it clear.

Thank you for this comment. We have added in the sample sizes for each data collection activity, to make it clearer, as follows:

This manuscript reports on Time 2 survey data (n=124), which were completed online in August 2018 by local program providers using an online survey tool (line 143).

Semi-structured telephone interviews (n=16) were undertaken with LPMs and LSC members from eight HKCC communities between August and October 2018 (Time 2; Time 1 interview data are reported elsewhere in an internal report (line 153).

Line 143 and line 153

The part "Discussion & conclusions" should be before the "Strengths & limitations"

We have taken your advice, and flipped these two sections.

Line 612-835

Line651 should be revised and find anothe way to say this ("While partner development was largely successful, some notable challenges were ev- 651
ident").

Thank you for this comment. We have revised the sentence to read:

Some notable challenges were evident with partner development.

Line 755

Reviewer 2 Report

Reviewer's report

Title: Exploring Factors Contributing to the Implementation of Ontario's Healthy Kids Community Challenge: Surveys and Key Stakeholder Interviews with Program Providers

Version: 1st version

Date: 04.10.2021

Reviewer's report:

The content of the research "Exploring Factors Contributing to the Implementation of Ontario's Healthy Kids Community Challenge" is of considerable interest, and this is a well-written manuscript. The authors mentioned all limitations transparently. However, I have a minor comment to improve the document.

The titles of the tables should be checked and corrected; there are some minor writing mistakes. e.g. in table 2, there is one bold "participants" written but not related to a sentence - the same in table 3. I would also recommend authors to visualise table 3, it would be much clear with major titles and subtitles, but it is their decision. 

Author Response

Reviewer #2 Feedback

The content of the research "Exploring Factors Contributing to the Implementation of Ontario's Healthy Kids Community Challenge" is of considerable interest, and this is a well-written manuscript. The authors mentioned all limitations transparently. However, I have a minor comment to improve the document.

Thank you for this feedback.

The titles of the tables should be checked and corrected; there are some minor writing mistakes. e.g. in table 2, there is one bold "participants" written but not related to a sentence - the same in table 3. I would also recommend authors to visualise table 3, it would be much clear with major titles and subtitles, but it is their decision. 

Thank you. We have revised the table headings, to be consistent throughout the document.